# Occurrence and Severity of Pain in Patients with Venous Leg Ulcers: A 12-Week Longitudinal Study

**DOI:** 10.3390/jcm9113399

**Published:** 2020-10-23

**Authors:** Paulina Mościcka, Justyna Cwajda-Białasik, Arkadiusz Jawień, Maciej Sopata, Maria T. Szewczyk

**Affiliations:** 1Department of Perioperative Nursing, Department of Surgical Nursing and Chronic Wound Care, Collegium Medicum in Bydgoszcz, Nicolaus Copernicus University in Torun, 85-821 Bydgoszcz, Poland; jcwajda@wp.pl (J.C.-B.); mszewczyk@cm.umk.pl (M.T.S.); 2Department of Vascular Surgery and Angiology, Collegium Medicum in Bydgoszcz, Nicolaus Copernicus University in Torun, 85-821 Bydgoszcz, Poland; ajwien@ceti.com.pl; 3Chronic Wound Treatment Laboratory, Chair and Department of Palliative Medicine, Hospice Palium, K. Marcinkowski University of Medical Sciences, 61-245 Poznan, Poland; maciej.sopata@skpp.edu.pl

**Keywords:** healing, leg ulcers, pain, ulceration, VAS, venous insufficiency

## Abstract

Background: The aim of the study was to analyze the dynamics of pain severity and its predictors in a group of patients with chronic venous leg ulcers. Methods: A 12-week longitudinal study included 754 patients with chronic venous leg ulcers. Subjective severity of pain was measured at weekly intervals with an 11-point visual analogue scale (VAS). Results: A significant decrease in VAS scores has been observed throughout the entire analyzed period. Higher severity of pain during follow-up was independently predicted by the presence of pus and/or unpleasant smell from the ulceration during the first visit, as well as by the occurrence of posterior and/or circumferential ulcers. The presence of ulcer redness during the first visit was associated with lesser pain severity; also, a significant interaction effect between the ulceration redness and warmth was observed. Conclusions: Implementation of complex holistic care may contribute to a substantial decrease in the occurrence and severity of pain in a patient with venous leg ulcers. Pain control seems to depend primarily on clinical parameters and topography of venous ulcers. The predictors of pain severity identified in this study might be considered during the planning of tailored care for patients with venous leg ulcers.

## 1. Introduction

Leg ulcers are a widespread chronic condition which constitutes a significant challenge for healthcare systems. It is estimated that 1 per every 800 persons in the United States presents with leg ulcers [1], with the majority of the ulcerations, up to 70–90%, being a result of chronic venous insufficiency [2,3].

Pain is a common ailment reported by the vast majority of patients with leg ulcers [4,5]. In a large proportion of those patients, pain is chronic, persistent or intermittent. Pain may be associated with underlying disease (i.e., venous insufficiency), ulceration itself, its inflammation and/or infection, or be evoked by dressing change [6,7]. Usually, pain is more severe in the early stages of treatment and then attenuates due to the progress of the healing process and application of compression therapy. However, pain associated with chronic venous insufficiency may also reappear after complete healing of the ulcer. Exacerbation/recurrence of pain can also be a marker of ulcer infection or its inappropriate local treatment.

Pain associated with chronic leg ulcers was shown to have a profound effect on the activities of daily living and the quality of life of the patients, and because of its negative influence on therapeutic compliance was also reported as an unfavorable prognostic factor [8,9]. Therefore, leg ulcer pain should be regularly monitored in terms of its severity, extent, and character, and these data can be later used to develop a tailored treatment plan for each patient [10]. Although the prevalence of pain associated with chronic leg ulcers was the subject of many previous studies, except from a few projects [11,12], the research had no longitudinal character. Hence, it is still unclear if a straightforward relationship exists between the dynamics of ulcer pain, the natural history of the ulceration and therapeutic interventions. Moreover, it needs to be emphasized that pain is a subjective sensation, the severity of which is affected by a plethora of factors, both clinical and patient-related; the results of research on those factors are to a large degree inconclusive [13,14].

Determination of pain dynamics and its predictors in persons with chronic venous leg ulcers seems crucial for the development of tailored interventions aimed at the improvement of patients’ wellbeing, and hence, also prognosis. Therefore, the aim of this longitudinal study was to analyze the severity of pain and its predictors in a group of 754 patients with chronic venous leg ulcers treated at a leading Polish tertiary center. To the best of our knowledge, this is one of a few longitudinal studies dealing with the problem in question, involving the largest group of patients examined to this date.

## 2. Experimental Section

### 2.1. Patients

A 12-week longitudinal study included 754 patients with chronic venous leg ulcers, qualified for treatment at the Chronic Wound Treatment Unit of the University Hospital between January 2001 and June 2019. The patients were eligible for the study if they presented with the ankle-brachial index (ABI) values between 0.9 and 1.3, had the diagnosis of chronic venous insufficiency confirmed on Duplex scan and complete medical documentation till the end of the 12-week follow-up period or complete healing of the ulcer, whichever occurred first. All patients were treated according to current standards, involving causal treatment (compression therapy), local management, TIME strategy (T- Tissue, I-Infection, M- Moisture, E- Edges), education and tertiary prevention.

### 2.2. Ethical Aspects

The protocol of the study was approved by the Local Bioethics Committee, and written informed consent was sought from all the participants.

### 2.3. Analyzed Parameters

The analysis included the data from medical documentation collected on enrollment and during subsequent control visits scheduled at weekly intervals, among them sociodemographic characteristics of the participants, information about comorbidities, history of chronic venous insufficiency, history of current leg ulcer, location of the ulceration(s), depth, and number thereof. The ulceration depth was classified based on the degree of the skin involvement, with ulcerations involving solely the epidermis considered as “superficial” and those penetrating to the dermis considered as “deep.” The latter category included both ulcerations with partial involvement of the dermis and those that penetrated across the whole dermis thickness. All patients were assessed according to Clinical-Etiology-Anatomy-Pathophysiology (CEAP) classification for chronic venous disease on enrollment, with only C6 patients included in the study. The severity of pain during consecutive control visits was estimated with an 11-point visual analog scale (VAS) where 0 corresponded to the lack of pain and 10 to the most severe pain possible. Ulceration area (in square centimeters) was measured electronically at biweekly intervals with a Visitrac appliance. In patients with multiple ulcers, the area of the largest ulceration was considered during the analysis. The analysis also included other clinical characteristics of the ulceration, among them its warmth, redness with a diameter greater than 2 cm, swelling, presence of pus, unpleasant smell, and pain.

### 2.4. Statistical Analysis

Statistical analysis was carried out with a Statistica 10 package (StatSoft, Tulsa, OK, USA). Normal distribution of quantitative variables was verified with the Shapiro–Wilk test. Summary characteristics of quantitative variables are presented as descriptive statistics, i.e., arithmetic means, standard deviations, medians, lower and upper quartiles, minimum and maximum values. Statistical characteristics of qualitative variables are shown as numbers and percentages. Statistical significance of sociodemographic and clinical variables as the predictors of pain severity during the 12-week follow-up was verified on univariate and multivariate analysis of variance (ANOVA) for repeated measures. The multivariate ANOVA model included the variables which turned out to be significant predictors of pain severity (*p* ≤ 0.05) on univariate analysis.

## 3. Results

Sociodemographic and clinical characteristics of the study participants are summarized in Table 1.

A significant decrease in the subjective severity of pain has been observed throughout the entire analyzed period (Table 2, Figure 1a). At the baseline, mean severity of pain on an 11-point VAS was 5.86 pts and then decreased to 3.67 pts and 2.33 pts at 6 and 12 weeks of follow-up, respectively. At the timepoints mentioned above, the presence of pain (VAS > 0 pts) was declared by 99.5%, 86.6%, and 66.9% of the study participants, and pain with the severity of at least 5 pts was reported by 77.6%, 36.4%, and 18.8% of the patients, respectively. In 10 (1.3%) patients, ulcers healed entirely before the end of the 12-week follow-up period, at one week (*n* = 2), four (*n* = 4), six (*n* = 3), and eight weeks (*n* = 1), respectively.

Repeated measure ANOVA (analysis of variance) identified posterior and/or circumferential ulcers, presence of pus, unpleasant smell, and pain of the ulcer during the first visit and baseline ulceration area >8.25 cm^2^ (median value for the study group) as the significant predictors of higher pain severity throughout the follow-up period (Figure 1b–f). In turn, redness and warmth of the ulcer during the first visit turned out to be predictive factors of lesser pain severity during the study period (Figure 1g,h).

Sex and age of the patients (up to 65 years vs. more than 65 years), duration of underlying chronic venous insufficiency (up to 20 years vs. more than 20 years), the time elapsed since the development of present ulceration (up to 12 months vs. more than 12 months), comorbidities, overweight and/or obesity, presence of medial, lateral, or anterior ulcers, number of ulcer locations, wound depth (deep vs. superficial wounds), number of concomitant ulcers (single vs. multiple), and ulceration swelling during the first visit did not exert a significant effect on the severity of pain during the analyzed period (Table 3).

During the next stage of the analysis, we verified whether any of the significant predictive factors identified above was an independent predictor of higher pain severity. The analysis was limited solely to the objective factors, i.e., the presence of posterior and/or circumferential ulcers, baseline ulceration area > 8.25 cm^2^, and signs of wound inflammation (pus, unpleasant smell) during the first visit. The pain of the wound during the first visit was not considered in the analysis as a subjective factor being directly related to the dependent variable. As neither the presence of pus or unpleasant smell from the ulceration during the first visit did not reach a threshold of statistical significance when included in a single ANOVA model (pus: F = 1.26, *p* = 0.235, unpleasant smell: F = 0.72, *p* = 0.736), they were substituted by a single variable “Pus and/or Unpleasant Smell” which turned out to be a significant predictor of pain severity (F = 9.55, *p* < 0.001).

Multivariate ANOVA demonstrated that higher severity of pain during the analyzed period was independently predicted by the presence of pus and/or unpleasant smell from the ulceration during the first visit, as well as by the occurrence of posterior and/or circumferential ulcers (Table 4).

Another multivariate analysis showed that lesser severity of pain during the follow-up period was independently predicted by the presence of ulcer redness during the first visit, but not by the ulceration warmth. However, a significant “Redness*Warmth” interaction effect has been observed as well (Table 4).

## 4. Discussion

The 12-week longitudinal study showed that with the time elapsed since the onset of treatment at our center, subjective severity of pain reported by patients with venous leg ulcers has been decreasing significantly. The decrease in pain severity was more or less linear. On admission and after 12 weeks of follow-up, presence of pain (VAS > 0 pts) was declared by 99.5% and 66.9% of the study participants, respectively.

Evaluation of pain occurrence and severity, as well as the dynamics of these parameters in patients with venous leg ulcers can be challenging. Although those issues were a subject of many previous studies, the vast majority of them were carried out at a single timepoint, had cross-sectional character and suffered from an array of confounders associated with differences in ulceration characteristics, treatments, and pain measurement methods. Based on published data, the prevalence of pain in patients with venous leg ulcers seems to vary considerably, from less than 30% to more than 90% [14,15], and even more significant discrepancies exist with regards of the pain severity [3,15,16,17,18].

Slightly more information about the true scale of the problem originates from a few longitudinal studies in which, like in our current research, the severity of pain was measured at various timepoints during treatment. In a 12-week longitudinal study of 65 patients with venous leg ulcers conducted by Charles [11], the presence of pain measured with an 11-point VAS at the beginning and the end of follow-up was reported by 71% and only 10% of the participants, respectively; in 65% of the patients, the ulcers healed entirely during the follow-up period. Mean severity of pain at the baseline and the end of the study was 4.5 pts and 0.4 pts, respectively, with a statistically significant decrease (down to 1.5 pts on average) observed solely after the first two weeks of treatment [11]. In turn, Vandenkerkhof et al. [12] analyzed the dynamics of pain in 424 patients with venous and mixed leg ulcers. Presence of pain at the baseline and after complete healing of the ulcer (the study endpoint) was declared by 82% and 32% of the patients, respectively, with the median severity on an 11-point VAS scale equal to 2 pts and 0 pts, respectively [12]. Thus, both our experiences and the results quoted above imply that even considering discrepancies associated with potential confounders (including baseline VAS scores), holistic care might contribute to a substantial decrease in the occurrence and severity of pain experienced by patients with venous leg ulcers.

The analysis of descriptive statistics for consecutive control visits demonstrated that the subjective severity of pain in some patients was higher and decreased at a slower rate than in others. These observations were confirmed by the results of repeated measures ANOVA. Two of the factors identified on univariate analysis (posterior and/or circumferential ulcer and presence of pus and/or unpleasant smell) turned out to be independent predictors of higher pain severity during the 12-week follow-up. These findings seem logical in light of our current knowledge of venous ulcer biology. Because of anatomical conditions, both posterior and circumferential ulcers are more prone to contact with various surfaces, e.g., during sitting and lying down, and hence, more exposed to pain stimuli. Not surprisingly, regardless of the ulcer location, more severe pain was also associated with the presence of pus and/or unpleasant smell from the ulceration, as both these signs point to a likely presence of bacterial wound infection; the latter is usually associated with impaired healing and exacerbation of pain.

Moreover, our analysis identified redness of the ulcer at the baseline as a factor associated with lesser severity of pain during the follow-up. ANOVA demonstrated both the main effect of the “Redness” variable and the “Redness*Warmth” interaction effect on the severity of pain. Redness of the ulceration, in some cases accompanied by its excessive warmth, might be a sign of hyperemia, an established factor facilitating wound healing, and hence, contributing to a lesser severity of pain. However, it should be stressed that inflammation is by definition the combination of redness, warmth, swelling, and pain. Surprisingly, pain/tenderness of the ulcer during the first visit was not associated with subjective severity of pain throughout the study period even though it is one of the four cardinal signs of inflammation. 

Previous studies identified a number of factors modulating the severity of pain experienced by patients with venous leg ulcers, among them the degree of venous insufficiency [19] and its duration [20,21], ulceration area [14,20,21,22,23,24,25,26], number of concomitant ulcers [17], ulcer duration [26,27,28], presence of wound infection [29], concomitant arthritis [30], wound and pain management strategies [12,14], as well as patient sex [12,17,19,26,31,32], marital status [30], financial condition [1], overall quality of life [12], and even the time of the day and the season of the year when the pain was measured [26,33,34,35]. Noticeably, published data on the influence of many of those factors, especially sociodemographic variables, on the severity of pain in patients with venous leg ulcers are inconclusive [13,14]. Some of the factors mentioned above were also included in our analysis but were not identified as significant predictors of pain severity. Thus, although a broadly defined patient disposition (sex, social factors, psychological status) with no doubt plays an important role in pain control, the latter seems to depend primarily on clinical parameters and topography of the ulceration.

The independent predictors of higher pain severity identified in this study might find application in clinical practice. Identification of patients with venous leg ulcers who are more likely to experience persistent pain during treatment might be vital for the therapeutic outcomes. Previous studies confirmed unequivocally that ulcer pain jeopardizes significantly the activities of daily living and social contacts of the patients [3,8,16,19,23,33,36,37,38,39], deteriorates the quality of their life [3,22,30,33,35,37,40,41,42,43,44,45,46,47,48] and can even be a determinant of worse treatment compliance [49]. Therefore, the patients with unfavorable baseline profiles of risk factors, more prone to experience persistent pain, could be offered a complex therapeutic program, including also psychological intervention and consultations of a pain management specialist [50]. Moreover, a higher frequency of scheduled control visits could be considered in such patients to boost their treatment compliance.

### Study Strengths and Limitations

An unquestioned strength of this study is the large sample size and complex approach to the problem of pain severity determinants during a long-term follow-up. However, the fact that the study group included persons with a history of previous and/or persistent ulcers might constitute a potential drawback; according to literature, individuals who have been experiencing pain for a longer time tend to assess its severity differently than those in whom pain appeared more recently [1]. Although we did not demonstrate a significant effect of ulceration history/duration on the severity of pain, it cannot be excluded that patients who developed ulcers earlier assessed otherwise subjective severity of pain according to different criteria than persons with the first and/or recently developed wound. Moreover, it needs to be stressed that the study included patients recruited over a relatively long period of time, between 2001 and 2019; while all patients were treated according to current standards, it cannot be excluded that some minor changes in practice or individual variation in care might have an impact on the study results.

## 5. Conclusions

Pain constitutes a significant problem in patients with venous leg ulcers. Implementation of complex holistic care seems to contribute to a substantial decrease in the occurrence and severity of pain. Presence of posterior and/or circumferential ulcers and pus and/or unpleasant smell from the ulceration are independent predictors of higher pain severity. These factors should be considered during the planning of complex tailored care for patients with venous leg ulcers.

## Figures and Tables

**Figure 1 jcm-09-03399-f001:**
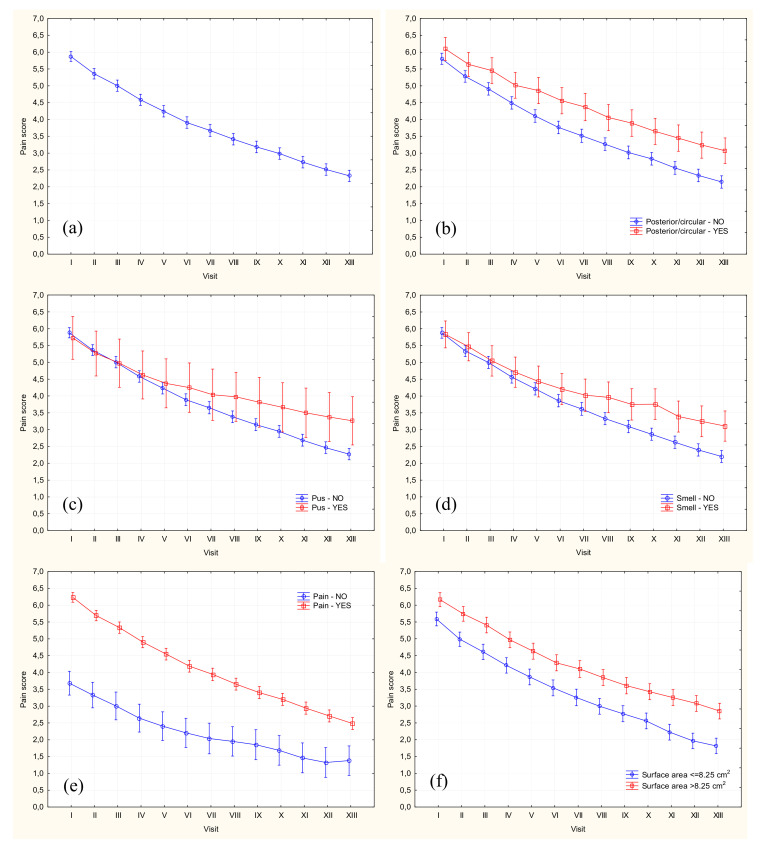
Mean pain scores (±95% confidence intervals) overall (**a**) and according to ulcer characteristics: presence of posterior/circumferential ulceration (**b**), presence of pus (**c**), unpleasant smell (**d**) and/or pain/tenderness during the first visit (**e**), and surface area at the baseline (**f**). Visit I denotes enrollment visit at Week 0 and visit XIII the last visit at week 12.

**Table 1 jcm-09-03399-t001:** Characteristics of the study group included in the analysis.

Parameter	Value
Women	*n* = 485 (64.3%)
Mean (±SD) age (years)	65.7 ± 12.09
Age > 65 years	*n* = 415 (55.0%)
Median (range) duration of underlying disease (years)	24 (0–70)
Underlying disease > 20 years	*n* = 392 (52.0%)
Median (range) duration of current ulcer (months)	12 (1–504)
Duration of current ulcer > 12 months	*n* = 296 (39.3%)
Comorbidities	*n* = 647 (85.8%)
Rheumatoid arthritis	*n* = 118 (15.6%)
Arthritis	*n* = 234 (31.0%)
Diabetes mellitus	*n* = 160 (21.2%)
Atherosclerosis ^1^	*n* = 96 (12.7%)
Cardiovascular disease ^2^	*n* = 184 (24.4%)
Overweight/obesity (BMI ≥ 25 kg/m^2^)	*n* = 633 (84.0%)
Obesity (BMI ≥ 30 kg/m^2^)	*n* = 354 (46.9%)
Medial ulceration	*n* = 469 (62.2%)
Posterior ulceration	*n* = 121 (16.0%)
Anterior ulceration	*n* = 129 (17.1%)
Lateral ulceration	*n* = 216 (28.6%)
Circumferential ulceration	*n* = 24 (3.2%)
Posterior/circumferential ulceration	*n* = 141 (18.7%)
Ulcer locations ≥ 3	*n* = 58 (7.7%)
Multiple ulcerations	*n* = 357 (47.3%)
Deep ulceration	*n* = 659 (87.4%)
Median (range) ulceration area at the baseline (cm^2^)	8.25 (0.12–538)
Baseline ulceration area > 8.25 cm^2^	*n* = 373 (49.5%)
Pus	*n* = 40 (5.3%)
Unpleasant smell	*n* = 103 (13.7%)
Redness	*n* = 472 (62.6%)
Swelling	*n* = 124 (16.4%)
Warmth	*n* = 369 (48.9%)
Pain	*n* = 646 (85.7%)

BMI—body mass index, SD—standard deviation. ^1^ Based on ABI values, patients with lower limb atherosclerosis were excluded from the study. ^2^ Patients with cardiac manifestations (according to NYHA or a history myocardial infarction) and/or cerebral manifestations (a history of stroke/TIA).

**Table 2 jcm-09-03399-t002:** Statistical characteristics of pain severity during consecutive control visits.

Week	*n*	VAS 0	VAS 5+	Mean ± SD	Median	Quartiles	Range
0	754	4 (0.53%)	585 (77.59%)	5.86 ± 2.05	6.0	5.0–7.5	0–10
1	752	22 (2.93%)	527 (70.08%)	5.35 ± 2.14	5.0	4.0–7.0	0–10
2	752	34 (4.52%)	440 (58.51%)	5.00 ± 2.30	5.0	3.5–6.5	0–10
3	752	46 (6.12%)	391 (51.99%)	4.58 ± 2.29	5.0	3.0-6.0	0–10
4	748	60 (8.02%)	334 (44.65%)	4.23 ± 2.34	4.5	2.5–6.0	0–10
5	748	80 (10.70%)	298 (39.84%)	3.90 ± 2.36	4.0	2.0–5.5	0–10
6	745	100 (13.42%)	271 (36.38%)	3.67 ± 2.46	3.5	2.0–5.5	0–10
7	745	110 (14.77%)	238 (31.95%)	3.41 ± 2.34	3.5	1.5–5.0	0–10
8	744	137 (18.41%)	217 (29.17%)	3.19 ± 2.39	3.0	1.0–5.0	0–9
9	744	154 (20.70%)	193 (25.94%)	2.99 ± 2.35	3.0	1.0–5.0	0–9
10	744	194 (26.08%)	164 (22.04%)	2.73 ± 2.37	2.5	0.0–4.0	0–10
11	744	216 (29.03%)	164 (22.04%)	2.51 ± 2.35	2.0	0.0–4.0	0–9
12	744	246 (33.06%)	140 (18.82%)	2.33 ± 2.31	2.0	0.0–4.0	0–9

SD—standard deviation, VAS—visual analog scale.

**Table 3 jcm-09-03399-t003:** Demographic and clinical factors as the predictors of pain severity; the results of univariate analysis.

Predictor	SS	df	MS	F	*p*
Follow-up time	11261.4	12	938.4	672.18	<0.001
Age > 65 years	23.3	12	1.9	1.39	0.163
Female sex	28.5	12	2.4	1.70	0.059
Underlying disease > 20 years	14.6	12	1.2	0.87	0.578
Current ulcer > 12 months	24.2	12	2.0	1.44	0.137
Comorbidities	27.6	12	2.3	1.65	0.071
Overweight/obesity	13.23	12	1.1	0.79	0.662
Obesity	23.4	12	2.0	1.40	0.159
Medial ulceration	27.0	12	2.3	1.62	0.079
Posterior ulceration	47.47	12	3.96	2.85	0.001
Anterior ulceration	4.64	12	0.39	0.28	0.993
Lateral ulceration	9.7	12	0.8	0.58	0.859
Circumferential ulceration	36.32	12	3.03	2.18	0.010
Posterior/circumferential ulceration	92.84	12	7.74	5.09	<0.001
≥3 ulcer locations	7.96	12	0.66	0.48	0.930
Deep ulceration	9.86	12	0.82	0.59	0.852
Multiple ulcers	18.3	12	1.5	1.09	0.361
Pus	72.85	12	6.07	4.37	<0.001
Unpleasant smell	123.31	12	10.28	7.42	<0.001
Redness	40.3	12	3.4	2.41	0.004
Swelling	8.82	12	0.74	0.53	0.899
Warmth	115.9	12	9.7	6.99	<0.001
Pain	215.83	12	17.99	13.09	<0.001
Ulceration area > 8.25 cm^2^	46.2	12	3.8	2.76	0.001

df—degrees of freedom, F—value of F-statistic, MS—mean sum of squares, SS—sum of squares.

**Table 4 jcm-09-03399-t004:** Ulcer characteristics as the predictors of higher and lesser pain severity; the results of multivariate analysis.

Predictor	SS	df	MS	F	*p*
Higher pain severity					
Pus/unpleasant smell	106.09	12	8.84	6.44	<0.001
Posterior/circumferential ulceration	38.71	12	3.23	2.35	0.005
Ulceration area > 8.25 cm^2^	20.42	12	1.70	1.24	0.248
Lesser pain severity					
Redness	56.08	12	4.67	3.41	<0.001
Warmth	24.30	12	2.03	1.48	0.124
Redness*Warmth	133.50	12	11.13	8.12	<0.001

df—degrees of freedom, F—value of F-statistic, MS—mean sum of squares, SS—sum of squares.

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
