# Peer review of "Occurrence and Severity of Pain in Patients with Venous Leg Ulcers: A 12-Week Longitudinal Study"

_jcm, 2020, doi:10.3390/jcm9113399_

Round 1
Reviewer 1 Report
This paper reports on an issue that is often not considered when caring for people with chronic venous leg ulcers. Relevant background context is provided.
The design is appropriate and the authors are commended for undertaking such a follow up design in the population of interest. A major strength of the study is the large sample size and that data were collected over time. It is interesting that it appears that the study was conducted from data spanning 18 years eg 2001 to 2019. I wonder if changes in practice over time would have had any impact on the results. Relevant statistics were applied to the data. As noted by the authors it may be interesting to continue to follow up this line of investigations re the impact of pain and redness in the ulcer. If this can be replicated it may well provide clearer guidelines for treating patients in the early stages of their treatment. Also it would appear that in future studies the area of wound would be an important variable to consider.
Author Response
- It is interesting that it appears that the study was conducted from data spanning 18 years eg 2001 to 2019. I wonder if changes in practice over time would have had any impact on the results.
Re: Since our center has been established in 2000, we have been treating all patients according to global standards, combining causal approach (compression therapy) with local management, TIME strategy, education and tertiary prevention. Consequently, no major changes in practice occurred despite a relatively long analyzed period. This has been emphasized in the revised Methods section. Nevertheless, we also mentioned minor changes in practice and individual variance in care as potential limitations of the study in the revised Discussion section.
Reviewer 2 Report
This is an interesting study that as acknowledged has been under researched but is of importance to people with venous leg ulcers. The article is well written but I just have some comments about some clarifications that may be of benefit to this article.
Methodology
It would be of benefit to explain how some of the parameters were determined i.e I initially thought that circular ulceration meant circumferential but then I wasn't sure what posterior/circular ulceration was. Venous leg ulcers are not usually circular so not sure if this is referring to the ulcer itself. How was deep ulceration measured?
Results
1.3% of ulcers healed in 12 weeks would be low compared to the literature and this should be commented on.
There is no discussion on medications that these people were on. Do you know if people were on pain medications and/or whether they were started on pain medications during the 12 weeks as this would impact on the study results.
I look forward to seeing the results of this large study published which will be of interest to health professionals caring for people with venous leg ulcers.
Author Response
- It would be of benefit to explain how some of the parameters were determined i.e I initially thought that circular ulceration meant circumferential but then I wasn't sure what posterior/circular ulceration was. Venous leg ulcers are not usually circular so not sure if this is referring to the ulcer itself.
Re: Indeed, we originally used a somehow vague term ‘circular’ instead of ‘circumferential’; this has been corrected throughout the revised manuscript. An ulcer was considered circumferential whenever it involved the whole lower leg, i.e. its anterior, posterior, medial and lateral aspects. For the purposes of statistical analysis, we created a ‘posterior/circular ulceration’ variable (‘posterior/circumferential ulceration in the revised manuscript) referring to ulcerations involving posterior aspect of the lower leg or whole lower leg (including its posterior aspect).
- How was deep ulceration measured?
Re: The ulceration depth was classified based on the degree of the skin involvement, with ulcerations involving solely the epidermis considered as ‘superficial’ and those penetrating to the dermis considered as ‘deep’. The latter category included both ulcerations with partial involvement of the dermis and those that penetrated across the whole dermis thickness. This has been specified in the revised Methods section.
- 1.3% of ulcers healed in 12 weeks would be low compared to the literature and this should be commented on.
Re: Probably, at least three various factors were involved. First, the study group varied considerably in terms of the ulceration area, with minimum and maximum values of 0.12 cm2 and 538 cm2, respectively. Further, nearly 50% of the patients had baseline ulceration area of 8.25 cm or more, with a relatively large proportion of subjects with extremely large ulcers that were less likely to heal completely. Similarly, the duration of current ulcer varied, from 1 month up to 504 months, and the proportion of patients who developed ulcers more than 12 months earlier approximated 40%. Also, in such cases, complete healing was less likely. Finally, a large proportion of patients treated with our center presented with recurrent ulcers, which was a subject of a previously published study (Moscicka et al. J Clin Nurs. 2016;25:1969-76, and some has the signs of inflammation/bacterial infection at the baseline.
- There is no discussion on medications that these people were on. Do you know if people were on pain medications and/or whether they were started on pain medications during the 12 weeks as this would impact on the study results.
Re: The vast majority of the patients (n=562, 74.5%) received pain medications, either non-steroid anti-inflammatory drugs (n=85.2%) or weak opioids (14.8%), throughout the study period. Most patients (54.1%) took the drugs once or twice a day. An additional statistical analysis did not demonstrate a significant effect of pain medication (either overall or specific type) on subjective severity of pain throughout the study period. This might be a consequence of the widespread pain medication use in the study group and/or intermittent character of the therapy (repeated measure ANOVA captures primarily the factors that produce a significant effect across the whole analyzed period).
Reviewer 3 Report
JCM-956421-occurrence and severity of pain in patients with venous leg ulcers Thank you for asking me to review this paper and in general I find it very interesting. However there are some major questions that do not seem to be answered in the text. Major points: Although this is a longitudinal study, the authors do not state that any point that I can see whether there is any uniformity or more importantly difference in the treatment of these patients. Were they all treated exactly the same? They all have exactly the same dressings and compression? Was there any difference in analgesia? Was there any difference in antibiotic usage? Without this information, then the results as currently presented cannot be taken as accurate. The authors need to report differences in treatment and do a sub- analysis of the different groups. I think this is particularly important in lines 195-200. The authors state that the redness and warmth are associated with lesser pain and explain this as being due to the hyperaemia of inflammation as a sign of healing. However there are two areas of concern with this. Firstly inflammation is by definition the combination of redness, warmth, swelling and pain. Therefore it is interesting that the authors concentrate only on the first two of these and do not find it surprising that pain is not associated even though it is one of the four cardinal signs of inflammation. Secondly, when doctors and nurses see redness and warmth associated with a venous leg ulcer, they almost always diagnose cellulitis and start antibiotics. The authors have not given any information as far as I can see as to whether there was any antibiotics given in any of these patients. This will obviously be a confounding variable. Another major problem is that the authors are misusing the CEAP classification. The CEAP classification should only be used as a description at the initial assessment. It was not designed to be used as a scoring system to assess changes in a longitudinal study. In order to assess progression of symptoms, the VCSS should be used. However, I do not think the authors are actually using the CEAP classification although they are quoting it. The CEAP classification for a venous ulcer is C6. Somehow the authors are quoting a median CEAP of 12 with a range of 0-17. This is impossible with the CEAP system and suggest that the authors might actually be confusing this with the VCSS. In table 1, the authors describe the population but at no stage did they define what they are describing. This is particularly important in: "atherosclerosis" - how is this determined? Many of the patients will have atherosclerosis and so there must be some cut-off point as to when it is regarded as significant to report it here. There needs to be a definition for this. "Cardiovascular disease" - there are many different forms that this could take and so the authors need to determine what their definition is. Is this angina, previous heart attack, previous strke, claudication or other factors? What is included and what isn't included. There needs to be a definition for this. "Deep ulceration"-how is this determined as to what is deep and what is superficial? There needs to be a definition of this. Minor points: In the abstract, conclusions section: line 24: what is meant by "implementation of complex holistic"? Is there a word missing after holistic? Figure 1: There is a lack of labelling in this figure. One assumes that the top left is A and the bottom right is H, but it is unclear as to how the graphs progress whether they run vertically or horizontally. Therefore it is hard to know what the bottom left graph is. In particular, although the others are labelled to show what is being measured, this is merely the difference between "Pain - no" and "Pain - yes" even though both show a pain score on the Y axis. This does not really make sense.
Author Response
- Although this is a longitudinal study, the authors do not state that any point that I can see whether there is any uniformity or more importantly difference in the treatment of these patients. Were they all treated exactly the same? They all have exactly the same dressings and compression?
Re: Since our center has been established in 2000, we have been treating all patients according to global standards, combining causal approach (compression therapy) with local management, TIME strategy, education and tertiary prevention. Consequently, no major changes in practice occurred despite a relatively long analyzed period. This has been emphasized in the revised Methods section. Nevertheless, we also mentioned minor changes in practice and individual variance in care as potential limitations of the study in the revised Discussion section.
- Was there any difference in analgesia?
Re: The vast majority of the patients (n=562, 74.5%) received pain medications, either non-steroid anti-inflammatory drugs (n=85.2%) or weak opioids (14.8%), throughout the study period. Most patients (54.1%) took the drugs once or twice a day. An additional statistical analysis did not demonstrate a significant effect of pain medication (either overall or specific type) on subjective severity of pain throughout the study period. This might be a consequence of the widespread pain medication use in the study group and/or intermittent character of the therapy (repeated measure ANOVA captures primarily the factors that produce a significant effect across the whole analyzed period).
- Was there any difference in antibiotic usage?
Re: In line with the standards of care at our center, targeted antibiotic therapy was implemented in patients in whom the signs of inflammation co-existed with a positive result of microbiological testing for Escherichia coli, Staphylococcus aureus or Pseudomonas aeruginosa. An additional statistical analysis did not demonstrate a significant effect of either antibiotic therapy or presence of any of the pathogens mentioned above on subjective severity of pain throughout the study period, which again might be an inherent to the use of repeated measure ANOVA.
- The authors state that the redness and warmth are associated with lesser pain and explain this as being due to the hyperaemia of inflammation as a sign of healing. However there are two areas of concern with this. Firstly inflammation is by definition the combination of redness, warmth, swelling and pain. Therefore it is interesting that the authors concentrate only on the first two of these and do not find it surprising that pain is not associated even though it is one of the four cardinal signs of inflammation. Secondly, when doctors and nurses see redness and warmth associated with a venous leg ulcer, they almost always diagnose cellulitis and start antibiotics. The authors have not given any information as far as I can see as to whether there was any antibiotics given in any of these patients. This will obviously be a confounding variable.
Re: Thank you for this important comment; we have considered your remarks in the revised Discussion.
- Another major problem is that the authors are misusing the CEAP classification. The CEAP classification should only be used as a description at the initial assessment. It was not designed to be used as a scoring system to assess changes in a longitudinal study. In order to assess progression of symptoms, the VCSS should be used. However, I do not think the authors are actually using the CEAP classification although they are quoting it. The CEAP classification for a venous ulcer is C6. Somehow the authors are quoting a median CEAP of 12 with a range of 0-17. This is impossible with the CEAP system and suggest that the authors might actually be confusing this with the VCSS.
Re: You are absolutely right; the use of the classification system was inappropriate. Hence, all references to the CEAP system were deleted from the revised Results section and correct information was included in the revised Methods.
- In table 1, the authors describe the population but at no stage did they define what they are describing. This is particularly important in: "atherosclerosis" - how is this determined? Many of the patients will have atherosclerosis and so there must be some cut-off point as to when it is regarded as significant to report it here. There needs to be a definition for this. "Cardiovascular disease" - there are many different forms that this could take and so the authors need to determine what their definition is. Is this angina, previous heart attack, previous strke, claudication or other factors? What is included and what isn't included. There needs to be a definition for this.
Re: In the revised manuscript, all relevant terms have been defined in the Table’s footnote.
- "Deep ulceration"-how is this determined as to what is deep and what is superficial? There needs to be a definition of this.
Re: The ulceration depth was classified based on the degree of the skin involvement, with ulcerations involving solely epidermis considered as ‘superficial’ and those penetrating to the dermis considered as ‘deep’. The latter category included both ulcerations with partial involvement of the dermis and those that penetrated across the whole dermis thickness. This has been specified in the revised Methods section.
- In the abstract, conclusions section: line 24: what is meant by "implementation of complex holistic"? Is there a word missing after holistic?
Re: This has been corrected in the revised version; thank you!
- Figure 1: There is a lack of labelling in this figure. One assumes that the top left is A and the bottom right is H, but it is unclear as to how the graphs progress whether they run vertically or horizontally. Therefore it is hard to know what the bottom left graph is.
Re: The elements of the compound figure have been labeled accordingly and the legend has been expanded.
- In particular, although the others are labelled to show what is being measured, this is merely the difference between "Pain - no" and "Pain - yes" even though both show a pain score on the Y axis. This does not really make sense.
Re: The categories ‘Pain – NO’ and ‘Pain – YES’ refer to the lack of pain/tenderness or presence thereof during the first visit, respectively. This has been specified in the revised figure legend. During statistical analysis, we verified whether this variable had an effect on subjective severity of pain throughout the study period.